# Thyroid Hormone Disruptors Interfere with Molecular Pathways of Eye Development and Function in Zebrafish

**DOI:** 10.3390/ijms20071543

**Published:** 2019-03-27

**Authors:** Lisa Baumann, Helmut Segner, Albert Ros, Dries Knapen, Lucia Vergauwen

**Affiliations:** 1Centre for Organismal Studies, Aquatic Ecology and Toxicology, University of Heidelberg, Im Neuenheimer Feld 504, 69120 Heidelberg, Germany; 2Vetsuisse Faculty, Centre for Fish and Wildlife Health, University of Bern, Länggassstrasse 122, 3012 Bern, Switzerland; helmut.segner@vetsuisse.unibe.ch; 3Fischereiforschungsstelle LAZBW, Argenweg 50/1, 88085 Langenargen, Germany; afhros@gmail.com; 4Department of Veterinary Sciences, Veterinary Physiology and Biochemistry, Zebrafishlab, University of Antwerp, Universiteitsplein 1, 2160 Wilrijk, Belgium; dries.knapen@uantwerpen.be; 5Department of Biology, Systemic Physiological and Ecotoxicological Research (SPHERE), University of Antwerp, Groenenborgerlaan 171, 2020 Antwerp, Belgium; lucia.vergauwen@uantwerpen.be

**Keywords:** PTU, TBBPA, mode of action, pathways of toxicity, microarray, transcriptome analyses

## Abstract

The effects of thyroid hormone disrupting chemicals (THDCs) on eye development of zebrafish were investigated. We expected THDC exposure to cause transcriptional changes of vision-related genes, which find their phenotypic anchoring in eye malformations and dysfunction, as observed in our previous studies. Zebrafish were exposed from 0 to 5 days post fertilization (dpf) to either propylthiouracil (PTU), a thyroid hormone synthesis inhibitor, or tetrabromobisphenol-A (TBBPA), which interacts with thyroid hormone receptors. Full genome microarray analyses of RNA isolated from eye tissue revealed that the number of affected transcripts was substantially higher in PTU- than in TBBPA-treated larvae. However, multiple components of phototransduction (e.g., phosphodiesterase, opsins) were responsive to both THDC exposures. Yet, the response pattern for the gene ontology (GO)-class “sensory perception” differed between treatments, with over 90% down-regulation in PTU-exposed fish, compared to over 80% up-regulation in TBBPA-exposed fish. Additionally, the reversibility of effects after recovery in clean water for three days was investigated. Transcriptional patterns in the eyes were still altered and partly overlapped between 5 and 8 dpf, showing that no full recovery occurred within the time period investigated. However, pathways involved in repair mechanisms were significantly upregulated, which indicates activation of regeneration processes.

## 1. Introduction

During the last years, research on the effects of endocrine disrupting chemicals (EDCs) on fish has mainly focused on the disruption of the sex steroid hormone system, which can result in reproductive impairment of exposed organisms [1,2]. While it is also well documented that environmental chemicals can disrupt the thyroid hormone (TH) system of different species [3,4], the specific adverse outcomes that are caused by exposure to thyroid hormone disrupting chemicals (THDCs) in fish are less well described. Due to the essential role of THs in the regulation of multiple physiological and developmental processes [5], disturbances of these systems can impact diverse fitness traits of the organism. An important TH-regulated process is the metamorphosis of lower vertebrates, including craniofacial morphogenesis and eye development [6,7]. For instance, differentiation of cone photoreceptors and color vision is directly regulated by TH signaling, probably through TH receptor beta, which is expressed in the outer nuclear layer of the retina [8,9,10]. Consequently, these developmental processes are potential target systems for the toxicological effects of THDCs, and, in fact, some studies have already demonstrated that exposure of developing fish to THDCs results in craniofacial and eye pathologies [11,12,13]. Likewise, it has been documented that knockdown of specific thyroid-regulating genes can lead to reduced eye size and malformations of eyes of zebrafish larvae [14,15,16], which confirms the interaction between the TH system and eye development.

In recently published data from our own research, we exposed zebrafish embryos to THDCs with different modes of action: propylthiouracil (PTU), a pharmaceutical that inhibits TH synthesis [17], and tetrabromobisphenol-A (TBBPA), a flame retardant that has been reported to interact with TH receptors [18,19]. Moreover, the interaction of TBBPA with binding proteins and hepatic clearance have been reported [4], as well as competitive binding to the plasma-transporter protein of T4, transthyretin [18]. The pharmaceutical PTU was selected as a typical positive reference substance, while TBBPA is environmentally relevant with concentrations of up to 2.6 μg kg^−1^ dry weight in sediment samples and up to 1.2 μg kg^−1^ wet weight in fish [19]. Measurements from different English lakes revealed that concentrations of TBBPA range from 140 to 3200 pg L^−1^ (water), 330 to 3800 pg g^−1^ dry weight (sediment), and <0.29 to 1.7 ng g^−1^ lipid weight (fish) [20]. Even though TBBPA has a relatively short half-life and low potential to bioaccumulate [21], it has also been detected in human breast milk and plasma with concentrations up to 37.3 μg/kg lipid weight [22].

In our previous study, we showed that exposure of zebrafish embryos to these two different THDCs resulted in congruent eye pathologies, such as smaller eye size, altered cellular structure and reduced pigmentation of the retina [23]. The observed eye pathologies translated into impaired eye function, as indicated from altered optokinetic responses and light preferences of THDC-treated larvae. Gene expression analyses in whole-body homogenates of exposed larvae further revealed compound-specific transcript changes of multiple thyroid-related genes (thyroid receptors α and β, thyroperoxidase, deiodinase1, 2, and 3), that confirmed the different modes of action of the selected THDCs on the TH system. Multiple other studies have reported evidence of the thyroid-disrupting effects of these compounds in zebrafish larvae, including, for example, Van der Ven [17], who showed lowered TH levels in PTU-exposed zebrafish or Chan [24] who demonstrated changes in gene expression of thyroid-related genes in TBBPA-exposed zebrafish. Our experiments with these compounds indicated that different molecular changes in the TH system can lead to similar phenotypic adverse outcomes in eye development, e.g., reduced diameter and pigmentation of the retinal pigment epithelium, reduced eye size, and impaired optokinetic response. [23]. These observations are fully in line with a recently published study by Parsons et al. [25], who demonstrated direct impact of TBBPA on the TH system and crucial developmental processes of zebrafish embryos at similar concentrations. Consequently, we were further interested to learn if similar molecular pathways in the eyes of PTU- and TBBPA-exposed zebrafish were affected, even though the molecular initiating events of the THDCs differ. The exposure concentrations for both THDCs were chosen based on our previous study [23], as they showed comparable effect intensities on the measured endpoints for eye morphology and physiology and are in line with the concentrations used in the studies mentioned before.

Until now, investigations on specific molecular interactions of the eyes and TH system have been limited to the fundamental studies of Bagci [15] and Houbrechts [26], who used deiodinase knockdown in zebrafish. As a next step, we were interested to see how THDC exposure interferes with molecular pathways of eye development and function, to advance our understanding of TH regulation of fish eye development in an environmental context. Most importantly, in contrast to previous studies, the present experiments relied on tissue-specific analyses of isolated eye tissue to avoid dilution of gene expression signals of the target organ by other tissues. Similar to our previous study, we used PTU and TBBPA as the test chemicals and zebrafish as the experimental model, since it is well established for the study of eye development, morphology, physiology, and diseases [27,28]. We performed a full genome microarray analysis with subsequent analysis of affected molecular pathways in the eyes of 5 dpf (days post fertilization) THDC-exposed zebrafish larvae. The main focus of this approach was on identifying affected processes, pathways, and response patterns, not on specific genes and exact fold-changes, which is why no further validation of gene expression changes by qPCR was performed.

Additionally, we investigated the reversibility of transcriptional changes after a recovery period of 3 days in clean water until 8 dpf. This scenario is relevant in an environmental context, where short-term peak exposure is likely to occur, which is especially critical during early development. The results from this study provide new insights into the molecular pathways associated with adverse effects of TH disruption in developing fish, and, at the same time, advance our knowledge of the role of THs in the regulation of fish eye development.

## 2. Results

### 2.1. Overview of Total Numbers of Differentially Expressed Transcripts

Both treatments, as well as the subsequent recovery phase, had significant impact on transcript levels in the eyes of exposed larvae. From a total of 43,803 probes on the microarray, Figure 1 shows the numbers of transcripts that were significantly changed for each compound and time point (as well as the intersects between them).

Generally, the number of differentially expressed transcripts was much higher in PTU-treated fish than in TBBPA-treated ones (12× more at 5 dpf and 49× more at 8 dpf, see Figure 1). In both treatments, the number of differentially expressed transcripts decreased after the recovery period from 5 to 8 dpf. The decrease was about 30% for PTU and over 80% for TBBPA. The calculation of the intersects between each compound and time point revealed that a high number of differentially expressed transcripts was overlapping within each compound when comparing 5 to 8 dpf (over 50% of transcripts from 8 dpf were also differentially expressed at 5 dpf). Moreover, there was an overlap between the two compounds at both time points (over 50% of transcripts differentially expressed after TBBPA exposure were also differentially expressed after PTU exposure).

### 2.2. Descriptive Analysis: Biological Function of Up- or Down-Regulated Transcripts for Each Treatment Group

The proportion of either up- or down-regulated transcripts in the four different transcript lists (two compounds with two time points each) was calculated (Figure 2), resulting in eight transcript lists that were further analyzed. In PTU-treated fish at 5 dpf, the percentage of up-regulated transcripts was slightly lower (43.3%) than the percentage of down-regulated ones. This pattern changed significantly (Fisher’s exact test, *p* < 0.0001) after 3 days of recovery when the percentage of up-regulated transcripts was higher than the percentage of down-regulated transcripts. In contrast, in TBBPA-treated fish, the proportions of up- or down-regulated transcripts did not change (Fisher’s exact test, *p* = 1) between 5 and 8 dpf.

The three most enriched (significantly overrepresented, i.e., lowest p-values) gene ontology (GO) classes for each of the eight transcript lists were identified. Each list had a distinct response pattern with different GO classes involved. The top three enriched GO classes for each list are shown in the pie charts in Figure 2 and were related to the nervous/optic system (red), immune system (green), and molecular transformation processes (blue). This overview shows that the dominating GO-classes mostly differed between the eight lists. The most obvious similarities were found for PTU-treated fish in the down-regulated transcripts at 5 and 8 dpf where GOs related to the nervous/optic system were most enriched. Similar GOs were also found between PTU and TBBPA at 5 dpf, where the GOs response to external or biotic stimuli was, respectively, up-regulated. All other lists showed differing response patterns and no enriched GOs could be calculated for TBBPA 8 dpf (up-regulated).

### 2.3. Comparative Analysis: Biological Function of Transcripts with Distinct Expression Patterns Across Treatments

The list of all differentially expressed transcripts was used to perform a cluster analysis based on their expression pattern across treatments, resulting in five clusters (see Appendix A) that were subsequently analyzed in GOrilla to find enriched GO classes in each cluster. That analysis resulted in 14 most enriched GO classes (combined from all clusters), which were further sorted and visualized for their response pattern across treatments. The 14 GO classes were sorted in the same way for all treatments, by using PTU 5 dpf as a reference for the order of GO classes in Figure 3 (top: highest percentage of down-regulated transcripts, last: highest percentage of up-regulated transcripts).

In TBBPA-treated fish, the number of differentially expressed transcripts was much lower than for PTU-treated fish, even though the phenotypic effects in our previous study [23] were similar. Consequently, some of the GO classes were not represented in the differentially expressed transcripts, and the response pattern differed clearly from the one of PTU-treated fish. The most severely affected GO classes (= highest number of transcripts) for TBBPA at 5 dpf were GO classes involved in metabolic processes (similar to PTU), as well as proteolysis (down-regulated) and regeneration, immune system and catabolic processes (up-regulated). GO classes concerning metabolic processes were most prominently represented at 8 dpf.

The most obvious differences between the two compounds were, e.g., the response pattern for the GO class sensory perception, where at 5 dpf we found over 90% down-regulation in PTU-treated fish compared to over 80% up-regulation in TBBPA-treated fish. Another difference was found for the GO-class proteolysis, where the PTU treatment caused almost 70% up-regulation, whereas for TBBPA it was only 10%.

When comparing 5 and 8 dpf for each compound, it becomes obvious that the recovery period caused a change of the response pattern. In the PTU-treated group, the percentage of up-regulated transcripts increased for nearly all GO-classes, e.g., from almost 70% to over 90% in the GO-class proteolysis or from under 10% to over 40% in the GO-class sensory perception. In TBBPA-treated fish, the most obvious change was the general decrease of differentially expressed transcripts from 5 to 8 dpf, which makes the response pattern difficult to compare.

### 2.4. Specific Analysis: Effects on Vision-Related Transcripts

Additionally, selected GO classes were visualized as examples in heatmaps showing the different response patterns in the four contrast groups (see Appendix A) after the cluster analysis. The impact on GO classes related to phototransduction was particularly prominent in both the descriptive (Figure 2) and the comparative (Figure 3) analysis. In the first analysis, the GO classes nervous system process, signal transduction and response to light stimulus were identified. In the second analysis, within the GO class sensory perception, the sub-classes neurological system process, sensory perception of light and visual perception were most affected (based on the cluster analysis, from which only the highest level GO classes were shown, to increase clarity, see Appendix A).

In combination with the observed impairment of visual capacities in PTU- and TBBPA exposed zebrafish in our previous study [23], this led us to zoom in on these pathways related to phototransduction. Figure 4 gives an overview of the phototransduction and retinoid recycling pathways in the retina and how transcription levels of the different components were changed in the four treatment groups. PTU-treated fish showed a strong down-regulation of most phototransduction-related transcripts at 5 dpf. Typical genes include: opsins (light-sensitive proteins in photoreceptors), phosphodiesterase (regulator of signal transduction), and arrestin (regulator of signal transduction). Other differentially expressed eye-related transcripts not involved in phototransduction, but the morphology of the eye was, e.g., crystalline (transparent protein in cornea) or peripherin (stabilization protein in rods and cones). After depuration, the down-regulation in PTU fish was less strong, and some new transcripts came up that were up-regulated: rhodopsin, opsin, and phosphodiesterase. TBBPA-exposed fish showed less response in transcripts related to sensory perception: only arrestin, guanylate cyclase activator (part of G-protein signaling cascade), calcium binding protein (part of calcium cell signaling pathways), and phosphodiesterase were slightly down-regulated and opsins up-regulated at 5 dpf, whereas only a slight up-regulation of opsin at 8 dpf was detectable.

## 3. Discussion

Based on our previous work that revealed adverse changes in the development of eye structure and function in PTU- and TBBPA-exposed zebrafish larvae [23], the present study aimed to identify the underlying molecular pathways involved in these effects. To this end, full genome microarray analyses of RNA isolated from eye tissue of PTU- and TBBPA-exposed larvae were performed to detect affected processes, pathways, and response patterns. Additionally, we investigated if the transcriptional changes were reversible after 3 days of recovery in clean water until 8 dpf.

### 3.1. General Response Pattern

The first basic finding of our analyses was that PTU had a much stronger impact on transcriptional changes in the eyes of exposed fish than TBBPA, even though similar biological processes were affected. The number of differentially expressed transcripts was over 10-times higher for PTU, and the difference was even greater after the recovery period of 3 days. This was surprising, as morphological and physiological changes in the eyes were comparable in our previous study with the same exposure concentrations [23]. However, the differing extent of transcriptional response is probably due to the differing molecular initiating events of the two THDCs. The PTU response was much more specific and prominent, which probably results from its strong ability to lower TH levels as a specifically designed pharmaceutical. TBBPA instead, seemed to have more general, not only thyroid-related effects, which was evident from expression changes in various pathways, among which cell metabolism was one of the most affected. Thus, it could be assumed that the eyes were a less specific target organ for TBBPA. Yet, both compounds had comparable damaging/pathological effects in the eyes of exposed fish, as shown in our previous study at higher levels of biological organization [23]. Nevertheless, intersect analyses revealed that over 50% of differentially expressed transcripts after TBBPA exposure were also differentially expressed after PTU exposure (Figure 1). Apart from vision- and regeneration-related transcripts, which are discussed in the following paragraphs, we observed significant transcriptional changes in the GO classes hormone system and metabolism (see Appendix A), which were, again, much stronger in PTU-treated larvae, but common for both treatments. Concerning the GO class hormone-mediated signaling pathway, mainly down-regulation was observed for both compounds, but significantly more transcripts were affected by PTU at both time points (Figure 3 and Appendix A). For both, the number of affected transcripts was very low. Down-regulation of TRs was only found in the PTU recovery group, which at least partially confirms an inhibitory effect on hormone signaling in the eyes, which fits the observed adverse changes in eye development that are most probably caused by altered TH levels. As outlined in the introduction, THs are essential for the regulation of early vertebrate development, including the eye development (reviewed by Bhumika & Darras [29]) and consequently, disturbances of TH signaling will translate into adverse developmental changes, as observed in our previous study [23]. Our study extends this knowledge from mammalian species to the zebrafish model.

Additionally, many general metabolic processes were affected by the THDC treatment in the eyes of exposed zebrafish larvae, which is not surprising, as the primary function of THs is to regulate metabolism [30], which can be disturbed by exposure to THDCs (reviewed by Casals [31] and Jugan [32]). Specifically, GO classes related to nucleic acid metabolism seemed most affected, meaning that synthesis and degradation of DNA and RNA were influenced by the THDC exposure. Similar effects on purine transcript levels in heads of zebrafish larvae were described in our study by Bagci [15] after deiodinase knockdown in zebrafish embryos. It was hypothesized that purine metabolism and vision are tightly linked, since cGMP phosphodiesterase levels play an important role in phototransduction. In our study by De Wit [33] in which zebrafish liver transcriptomics were investigated after TBBPA exposure, we observed similarly affected pathways as in the present study. We found that TBBPA induced oxidative stress and general stress responses with numerous differentially expressed transcripts associated with defense mechanisms, metabolizing enzymes, and signal transduction. Again, these results fit well with the eye pathologies of THDC-exposed larvae we could demonstrate in our previous study [23].

### 3.2. Effects on Vision-Related Transcripts at 5 dpf

Despite the difference in the amounts of affected transcripts, our analyses of the biological functions revealed that similar processes in the eyes of exposed larvae were affected by the two THDC treatments, especially with regard to transcripts involved in neurological system, visual and sensory perception. However, the most obvious differences between the two compounds were, e.g., the response pattern for the GO class sensory perception, where at 5 dpf we found over 90% down-regulation in PTU-treated fish compared to over 80% up-regulation in TBBPA-treated fish. Independent of the type of approach for our data analyses, important transcriptional changes of neuro-, sensory-, and vision-related genes were always detected for both compounds and time points, as well as the different intersects. This effect was more prominent in PTU-exposed fish, but also considerable in TBBPA-treated fish. These results provide strong evidence that the treatment with both THDCs did induce molecular changes that affect the eye development and function of exposed zebrafish larvae, as already demonstrated in our previous study at higher levels of biological organization, i.e., cellular changes in the retina, reduced eye size, and impaired visual performance of PTU- and TBBPA-exposed larvae [23]. To understand these adverse effects at the molecular level, we were specifically interested in transcript changes of genes involved in phototransduction and retinoid acid recycling pathways in the eyes. As illustrated in Figure 4, almost all components of these pathways were differentially expressed due to the THDC treatment, which clearly implicates that the phototransduction physiology of exposed larvae was impaired. With PTU exposure, a strong down-regulation of almost all differentially expressed eye- and nervous system-related transcripts at 5 dpf was observed (Figure 4). As these are essential for morphology and physiology of the eyes, the observed down-regulation can most probably be linked to adverse effects on eye morphology and visual capacities of the exposed fish, as documented in our previous study at 5 dpf [23]. The transcriptional response pattern in TBBPA-treated fish differed from the one of PTU: fewer eye- and nervous system-related genes were differentially expressed after the treatment, and those that were affected were mainly up-regulated (opsins), which is surprising, as the observed effects at higher organization level were similar in the chosen exposure concentrations of PTU and TBBPA. However, as illustrated in Figure 4, essential components of the phototransduction pathway were also differentially expressed due to the TBBPA treatment, which confirms our previous findings on impaired visual capacities of exposed larvae.

Our analyses extend our previous findings and existing literature on TH regulation of eye development in vertebrates that can be disturbed by THDC exposure. It has been reported that in zebrafish, mice, and humans, THs regulate expression of opsins in the retina through the Thrb2 receptor [34]. Suzuki [35] described that the zebrafish retina has four different cone types, for which the relative amounts in the retina are regulated by THs. Similar work by Dong [11] showed that TRb mRNA expression is localized in two specific layers, the ganglion layer and the amacrine and/or bipolar cell layer of the developing retina of zebrafish, which was disrupted after exposure to the THDC 6-OH-BDE 47. This was recently supported by our research presented in Houbrechts [26], where we found reduced T3 levels and eye size in zebrafish with deiodinase knockdown. Similar findings have been reported in our study by Bagci [15], where we describe down-regulation of transcripts involved in purine metabolism, phototransduction, and nervous system/eye development in the head of deiodinase deficient zebrafish larvae. Deiodinases play an important role in the TH system, as they convert the inactive TH T4 to the active T3 form. PTU is known as a deiodinase inhibitor in mammals [36,37], but its inhibitory effect on fish deiodinases is less clear [36]. In our previous study [23], we observed up-regulation of dio2 and down-regulation of dio3 in whole body homogenate of PTU-treated zebrafish, which was not the case in the eye tissue of the present study. Similar observations were made for TR expression, for which both isoforms were down-regulated in the whole body of PTU-exposed larvae [23] but only in the recovery group of the present study (see Appendix A). Similarly, TBBPA treatment did not affect TR expression in the eyes (see Appendix A), while this has been demonstrated in whole zebrafish larvae before [23,37]. Despite the obvious differences between whole body and eye tissue measurements, these previously documented changes in TH signaling of the larvae are very likely to be involved in the observed changes of eye morphology, physiology, and vision-related pathways documented in our experiments with PTU and TBBPA.

A general neurotoxic mechanism of TBBPA might also play a role, as shown, e.g., by Chen [38], who exposed zebrafish embryos to TBBPA and observed neurobehavioral toxicity, together with transcriptional changes of pathways for neuronal development. A recent study by Park [39] showed that TBBPA is ototoxic in mice and zebrafish as it causes loss of zebrafish neuromasts and hair cells in the rat cochlea. The number of TBBPA studies that show similar results is increasing [40,41]. However, the described behavioral changes might partly be explained by the disrupting effect on eye development as observed in the present and in our recent study [23].

Consequently, even though the molecular initiating events in TH signaling may differ, the findings of the present study indicate that the toxicity pathways merge at a certain level to translate in uniform changes of eye development. Such observations are particularly relevant in the context of the current discussion about the adverse outcome pathway concept. Moreover, data from our own research [14,15,23,26] link TH disruption, either via knockdowns or via toxicological exposure, to morphological, physiological, and behavioral effects of the visual system, thus, effectively showing evidence of higher-level outcomes.

### 3.3. Evidence for Regeneration at 8 dpf

Interestingly, the response pattern of vision-related transcripts between 5 and 8 dpf clearly changed for both substances. While for TBBPA, only one up-regulated transcript (opsin) was still detectable at 8 dpf, the number of differentially expressed vision-related transcripts for PTU was still high, and the pattern changed from mainly down-regulated to more up-regulated at 8 dpf. Transcripts that were not affected at 5 dpf but became up-regulated after the depuration period were opsins (light-sensitive protein) and phosphodiesterase. Most of the transcripts that were down-regulated at 5 dpf, were still down-regulated but less apparent at 8 dpf. This change of response pattern for both substances provides evidence for activation of a recovery effect after the depuration period of 3 days, especially because transcripts from the GO class regeneration were strongly upregulated, which might be a compensatory response. Similar observations were made by Houbrechts [26], who observed recovery of adverse apical effects in the eyes of zebrafish with deiodinase knockdown at 7 dpf. This can be explained by the chosen method for the gene knockdown with morpholinos, that is known to be not fully persistent over longer periods.

Generally, we identified different GO classes that can be regarded as a basic response to regeneration/recovery in the eyes of exposed larvae. Genes involved in tissue development, immune system, cell redox homeostasis, and positive regulation of cellular component organization were differentially expressed for both compounds and time points, which is in line with the cellular and morphological changes in the eyes observed in our previous study [23]. These changes were even stronger at 8 dpf, which suggests that repair mechanisms were increasingly activated in the recovery phase after the THDC treatment was ceased. The same pattern was also observed for non-eye-related GO classes, and again, these changes were more prominent for PTU than for TBBPA.

The activation of repair mechanisms in the eyes of exposed larvae at 8 dpf might not only be a response to cellular changes in the eyes but could be enforced by different other factors. There are multiple studies that describe the involvement of THs in regeneration processes of fish, for example, Bouzaffour [42] showed that altered TH levels in methimazole-treated zebrafish resulted in impaired fin regeneration after amputation. Bhumika [43] demonstrated that lowered TH levels accelerate optic tectum re-innervation following optic nerve crush in zebrafish. Generally, THs seem to be crucial for different aspects of neuronal regeneration in vertebrates (reviewed by Bhumika & Darras [29]). In the present study, the strongest activation of genes related to regeneration was observed for suppressors of cytokine signaling (immunomodulation), major vault protein (mediates drug resistance), proto oncogene (helps to regulate cell growth and differentiation), heat shock proteins (prevention of protein damage under stress), and different activators of transcription. Moreover, multiple genes involved in tissue development and cell redox homeostasis were up-regulated. For instance, thioredoxin (redox protein, antioxidant), glutaredoxin (redox enzyme), and protein disulfide isomerase (catalyzes protein folding) suggest that the treatment caused oxidative stress and tissue damage in the eye, which was antagonized by those redox and repair proteins. For both THDCs, induction of oxidative stress and reactive oxygen species formation in fish and rodents has been described before [44,45].

Another aspect that has to be considered regarding the activation of repair mechanisms is the interplay of the thyroid and immune system. Concerning immune-related genes, PTU and TBBPA exposure resulted in very different response patterns (see Appendix A). TBBPA had almost no effect, with matrix metallopeptidase being the only affected immune-related gene. PTU-treated fish instead, showed transcriptional changes for 37 different immune-related genes at 5 dpf, of which the majority was up-regulated compared to the negative control. At 8 dpf, the number increased to 41 genes, and the percentage of up-regulated transcripts increased to over 90%. The strongest up-regulation was observed for chemokines (pro-inflammatory immune response), metallopeptidase (degradation of extracellular matrix), and complement factor properdin (tissue inflammation). The immunomodulatory effect of PTU in fish has been reported before [46,47,48], and the general influence of THs on immune functions of fish has been described as well [49]. In the present study, the stimulating effect of PTU on immune-related genes in the eyes of exposed zebrafish larvae might be an indicator for degenerative/pathological effects in the eye tissue that the immune system tried to compensate for. In fact, pathological alterations of retinal structures were identified in our previous study [23], which most probably result in the observed transcriptional changes of immune-related genes observed in the present study.

Our results show that within the 3-day-recovery period (after 5 days of THDC exposure), zebrafish larvae activated substantial repair and regeneration processes, associated with a general metabolic activation. At the same time, especially for PTU, many of the specific transcriptional responses that were detected after 5 days of exposure, were still present after the 3-day recovery phase (e.g., Appendix A: genes involved in hormone-mediated signaling and sensory perception), suggesting that the recovery process was not fully accomplished yet. Further research, with longer recovery periods and morphological/physiological assessments, is needed to investigate whether full recovery from disrupted eye development induced by early life exposure to THDC might be possible in the developing zebrafish larvae.

## 4. Materials and Methods

### 4.1. Ethics Statement

The EU Directive on the protection of animals used for scientific purposes (2010/63/EU) and the Commission Implementing Decision 2012/707/EU state that fish are non-protected animals until they are free feeding, i.e., 120 h post fertilization (hpf) for zebrafish [23]. All experiments of this study exceeding 120 hpf were approved by the Ethical Committee for Animals of the University of Antwerp (project ID 2015-51). Fish husbandry and all experiments were carried out in strict accordance with EU Directive 2010/63/EU [50].

### 4.2. Test Chemicals

Tetrabromobisphenol-A (TBBPA, CAS 79-94-7) and propylthiouracil (PTU, CAS 51-52-5) were purchased from Sigma Aldrich (St. Louis, MO, USA) with >98% purity. Exposure concentrations were 200 µg/L for TBBPA (0.37 µM) and 350 mg/L for PTU (2.04 mM). These sublethal concentrations were chosen based on our previously published study on TH disruption and eye development in zebrafish [23], as they showed comparable effects on eye morphology and physiology (specifically: the lowest observed effect concentrations for significant effects on reduced diameter and pigmentation of the retinal pigment epithelium, reduced eye size, impaired optokinetic response, and altered light-dark preference). TBBPA was dissolved in DMSO with a maximal final DMSO concentration in the exposure medium of 0.004%, which is below the limit of 0.01% set by OECD Test Guideline 236 [51]. Consequently, controls for the TBBPA treatment were also run with 0.004% DMSO. Both chemicals were applied in E3 embryo medium ([52], 5 mM NaCl, 0.17 mM KCl, 0.33 mM CaCl_2_, 0.33 mM MgSO_4_, pH 7.6). PTU was directly dissolved in it.

### 4.3. Zebrafish Maintenance and Exposure

Non-exposed adult wildtype zebrafish (*Danio rerio*) from an in-house laboratory colony at the University of Antwerp were used for egg production. They were kept in reconstituted freshwater with adjusted pH (using NaHCO_3_, 7.5 ± 0.3) and conductivity (using Instant Ocean^®^ Sea Salt, Blacksburg, VA, USA, 500 ± 15 μS/cm), a temperature of 28 ± 0.2 °C and a day/night cycle of 14/10 h. Water quality was monitored twice weekly, using Tetratest kits (Tetra Werke, Melle, Germany). Values for ammonium, nitrite, and nitrate were below 0.25, 0.3, and 12.5 mg/L, respectively. Adult zebrafish were fed four times per day, twice with granulated food (1.5% of their average weight, Biogran medium, Prodac International, Cittadella, Italy) and twice with frozen Chironomidae larvae, *Artemia* sp. nauplii, Chaoboridae larvae and *Daphnia* sp. (Aquaria Antwerp bvba, Aartselaar, Belgium) alternately. Four breeding pairs (two females and one male) were placed in breeding tanks with a perforated bottom for egg production. Eggs were collected early in the morning after the lights were switched on. Eggs were collected, rinsed, and transferred to clean reconstituted water 30 min after fertilization. All eggs were checked using a stereomicroscope (Leica S8APO, Leica Microsystems GmbH, Wetzlar, Germany) and only healthy, fertilized eggs were further used. Eggs from the different breeding pairs were mixed and randomly distributed into pre-incubated 24-well plates (= plates were filled with the solutions 1 day before the start of the exposure. Before the start of the test, solutions were renewed) containing the exposure solutions latest 2 h after fertilization (1 egg per well, 2 mL solution per well). Plates were incubated at 28.5 °C under a day/night cycle of 14/10 h. Exposure solutions were renewed daily with freshly prepared stock solutions to ensure constant chemical concentration levels and good water quality. Each exposure group was run in 6 biological replicates with 24 eggs per biological replicate/plate. After 5 days of exposure, 3 biological replicates of each treatment were sampled, resulting in 3 treatment groups which were used for the subsequent microarray analyses: control 5 dpf, PTU 5 dpf, and TBBPA 5 dpf. Larvae from the 3 remaining biological replicates of each treatment were carefully placed into new plates with clean water and raised until 8 dpf with daily water exchange. These larvae represent the 3 recovery groups for the microarray analyses: control 8 dpf, PTU 8 dpf, and TBBPA 8 dpf. The exposure set-up is summarized in Table 1. The choice of duration of the recovery period was based on our study by Houbrechts [26], in which we observed recovery from altered eye development at 7 dpf that was induced by dio-knockdown. This can be explained by the chosen method for the gene knockdown with morpholinos, that is known to be not fully persistent over longer periods. Based on this, we chose our exposure scenario with 5 + 3 days. Moreover, this took the short half-lifes (few hours) of our exposure compounds into account.

### 4.4. Sampling and RNA Isolation

To avoid confounding of treatment-related effects by circadian rhythms, sampling of all larvae was performed within maximum 3 h during the morning. For sampling of the eyes of the larvae, they were individually and consecutively transferred into ice-cold water for anesthesia. This approach was used to avoid the impact of anesthetics on subsequent transcriptomic analyses. Ice-cold water is an approved method for anesthesia of zebrafish larvae [53]. Single larvae were then placed under a stereomicroscope, and the eyeballs were dissected using a small syringe. The eyeballs were quickly transferred to a cooled collection tube on ice to avoid RNA degradation. Larvae were euthanized by decapitation. After collection of eyes from 10 individuals, the tube was frozen in liquid nitrogen and stored at −80 °C until further processing. Total RNA was isolated using the NucleoSpin RNA kit (Macherey-Nagel, Dueren, Germany), according to the manufacturer’s instructions. Standard laboratory precaution measures were taken to avoid RNAse contamination. RNA quality and quantity were determined with a Nanodrop spectrophotometer ND-1000 (Thermo Scientific, Waltham, MA, USA). RNA integrity (RIN) was verified using a bioanalyzer (Agilent 2100, Santa Clara, CA, USA). Only high-quality RNA (average RIN: 8.9 ± 1.0) was further processed for microarray analysis, which was the case for all samples.

### 4.5. cRNA Labelling and Hybridization

Total RNA was linearly amplified and labelled with the Low Input Quick Amplification Labelling Kit (Agilent Technologies) according to the manufacturer’s instructions. In summary, 100 ng RNA was reverse transcribed into cDNA using oligo dT primers. Afterwards, cDNA was transformed into cRNA and amplified. The cRNA of each sample was labelled once with Cy3-CTP and once with Cy5-CTP. Subsequently, the labelled cRNA samples were purified with the RNeasy mini spin column kit (Qiagen, Hilden, Germany). cRNA yield, quality, and dye incorporation efficiency were verified with the Nanodrop spectrophotometer. For final transcriptome analysis, we used Agilent’s Zebrafish Gene Expression Microarray V3 (AMADID 026437) in a 4 × 44k format, which is a full genome microarray containing 43,803 *Danio rerio* probes. For each of the two compounds, we used an *n* + 2 A-optimal design [54] where *n* is the number of samples, in this case, 12, and, thus, the number of arrays equals 14. 825 ng Cy3 and 825 ng Cy5 labelled and purified cRNA was hybridized on the microarrays for 17 h at 65 °C in a rotating (10 rpm) hybridization oven (Agilent Technologies). After hybridization, the slides were rinsed in Agilent wash buffers, acetonitrile and in stabilization/drying solution (Agilent Technologies) to wash and protect against ozone-induced dye-degradation. Microarray slides were scanned with a Genepix 4100A confocal scanner (Axon Instruments, Union City, CA, USA) at a resolution of 5 μm and excitation wavelengths of 635 nm and 532 nm in an ozone-free location (NoZone scanner enclosure, SciGene, Sunnyvale, CA, USA). Images were analyzed for spot identification and for quantification of fluorescent signal intensities with the Genepix Pro software 6.1 (Axon Instruments).

### 4.6. Analysis of Microarray Data

#### 4.6.1. Statistical Analysis

Quality control of the microarray data was performed with the R software package “arrayQualityMetrics” (version 3.34.0, [55]), and no outliers were detected. Spots were excluded from the analysis if FG < BG + 2SD (FG: foreground, BG: local background, SD: standard deviation of the local backgrounds of the entire array, [56]) for all arrays in the dataset. Background correction was done by applying a normal-exponential convolution model [57]. Within-array adjustment was performed using the Loess normalization [58], which is an intensity-dependent normalization of the red/green ratio. Responses in the eyes of larvae exposed until 5 dpf were contrasted to the corresponding controls at 5 dpf, and responses in the eyes of larvae allowed to recover until 8 dpf were contrasted to the corresponding controls at 8 dpf for the two compounds separately, using the R software package “Limma” (version 3.34.4, [59]) as described by Vergauwen [60]. After linear models were fitted to intensity ratios, an empirical Bayes method [58] was used to rank the probes in order of evidence of differential transcription. Differentially expressed transcripts were selected based on the following criteria: logFC (binary logarithm of the fold change) >0.75 (corresponding to a fold change of 1.68) and *p* < 0.05 where *p* is the adjusted *p* value after false discovery rate (FDR) control according to the Benjamini-Hochberg procedure [61]. A relatively low logFC cut-off was applied because the study was focused on identifying affected pathways rather than providing evidence for the involvement of individual genes. This resulted in 4 differentially expressed transcript lists: PTU 5 dpf, TBBPA 5 dpf, PTU 8 dpf, TBBPA 8 dpf.

Raw and analyzed microarray data, including the hybridization design, have been deposited in NCBI’s Gene Expression Omnibus (GEO, http://www.ncbi.nlm.nih.gov/geo) and are accessible through the GEO series accession number GSE121338 (reviewer only link: https://www.ncbi.nlm.nih.gov/geo/query/acc.cgi?acc=GSE121338, Enter token cfedagqgvnqjnyl into the box).

#### 4.6.2. Biological Interpretation

Clustering analysis of transcript changes was performed with the MultiExperiment Viewer software 4.8.1 (MeV, http://www.tm4.org/mev.html). Transcripts were further analyzed in GOrilla (http://cbl-gorilla.cs.technion.ac.il), which is a tool for identifying enriched (significantly overrepresented) gene ontology (GO) terms in lists of genes compared to a relevant background gene list [62].

(a) In a first descriptive analysis, for each compound and time point, a separate GO enrichment analysis was carried out for significantly upregulated transcripts on the one hand and significantly down-regulated transcripts on the other hand, using all differentially expressed transcripts as the background list. This allowed us to determine which GO classes were characteristic of each specific treatment relative to the full set of differentially expressed genes in any of the treatments. We chose not to use all transcripts on the array as the background list since this would imply that the selection of the genes represented on the array, made by the manufacturer, would affect our analysis. This resulted in 7 different lists of enriched GO classes (TBBPA 8 dpf up-regulated gave no result). The top 3 (highest *p*-value) enriched GO classes of each list were selected to describe the most important affected biological processes in each treatment (see Figure 2).

(b) In a second comparative analysis, all differentially expressed transcripts were used to perform a cluster analysis in MEV (k-means—Pearson correlation), resulting in 5 clusters (see Appendix A) containing transcripts with distinct expression patterns across treatments. These 5 clusters were subsequently analyzed in GOrilla to link biological functions to specific expression patterns. The transcript list of each cluster was set as target list and compared to a background list, again consisting of all differentially expressed transcripts, to determine how the specific cluster is distinct from the remaining clusters in terms of affected GO classes (see Appendix A, for lists of enriched GO classes in the 5 clusters). This analysis resulted in 14 GO classes significantly enriched in at least one of the clusters (i.e., expression patterns), meaning that these 14 GO classes are linked to specific expression patterns and could, therefore, be important for comparing the different treatments. These GO classes were then further studied for their response pattern across the different treatments. To facilitate easy interpretation, we visualized the relative number of up- versus down-regulated transcripts in the 14 GO classes for each compound and time point (see further in Figure 3).

## 5. Conclusions

In conclusion, our study clearly demonstrates (i) how specific eye-/vision-related transcripts were differentially expressed after the THDC exposure, (ii) that the transcript profiles differed at least partly according to the mode of action of the TCDs, (iii) and that the recovery time of 3 days was not sufficient to fully reverse those effects, even though regeneration processes were clearly activated. These results confirm existing literature that shows that photoreceptors, optic primordia, optic nerve development, and opsin expression in fish are under the influence of THs and can be disrupted by THDC treatment. To the best of our knowledge, full-genome transcriptomic analyses of isolated eye tissue of THDC-exposed fish have not been performed so far and aids in understanding the underlying molecular mechanisms that translate into impairment of visual capacities of exposed fish.

## Figures and Tables

**Figure 1 ijms-20-01543-f001:**
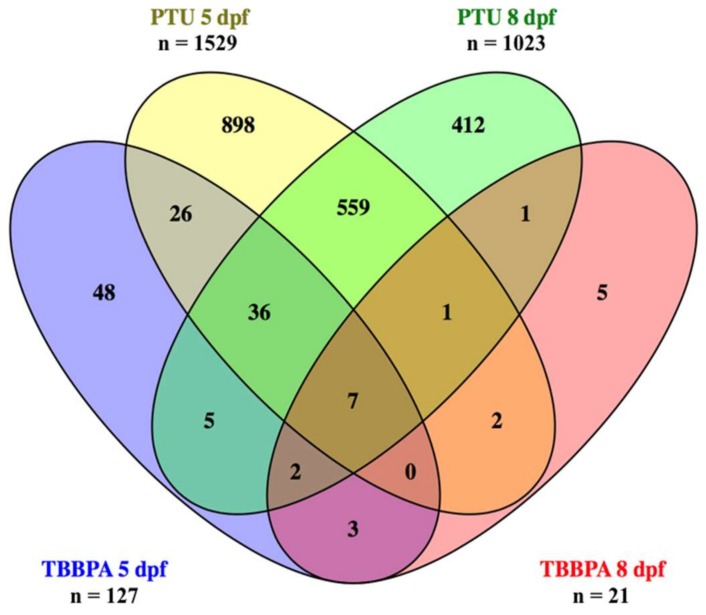
Number of differentially expressed transcripts in the different treatments and intersects between them. 5 dpf (days post fertilization) = continuous exposure from 0 to 5 dpf; 8 dpf = exposure until 5 dpf + 3 days of recovery.

**Figure 2 ijms-20-01543-f002:**
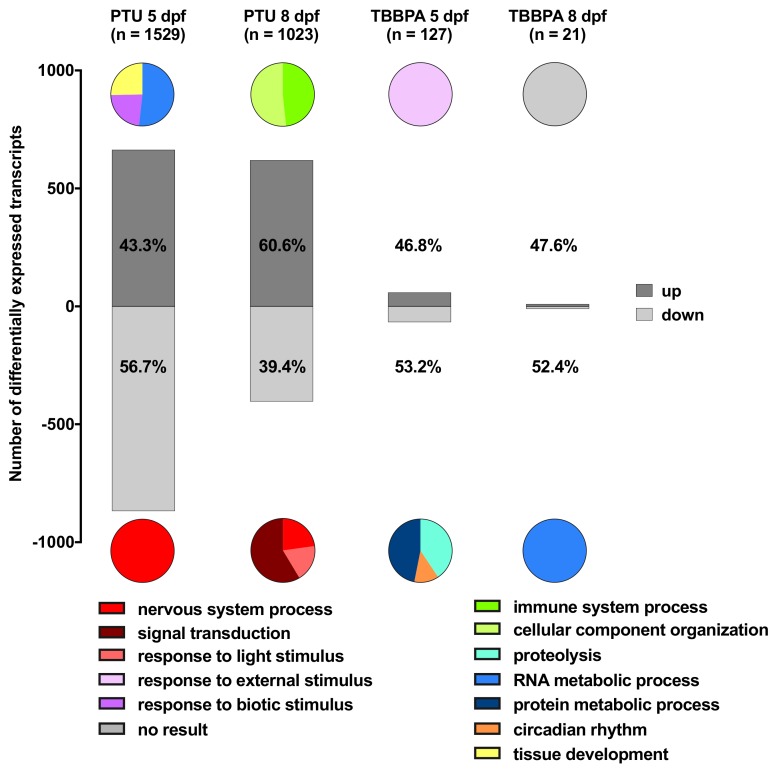
Transcriptional changes for eight different transcript lists (two compounds, two time points, up- or down-regulated) with respective enriched GO classes. The bars represent the numbers of up- or down-regulated transcripts for each list with the respective percentage written in it. The total number of differentially expressed transcripts of each list is referred to as “*n* =” on top of each bar. The upper pie charts show the top three enriched GO classes for the up-regulated transcripts of each list, the lower ones for the down-regulated transcripts (the complete pies are, thus, not representing the high total number of transcripts in each list, which would impede visualization).

**Figure 3 ijms-20-01543-f003:**
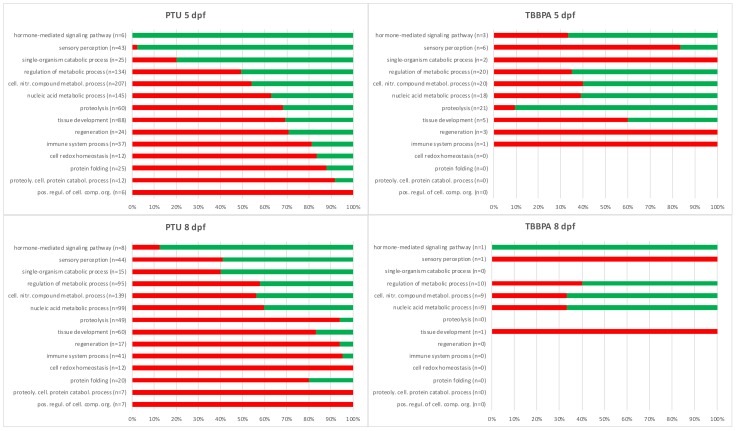
Overview of the percentage of up-regulated (red) or down-regulated (green) transcripts in the 14 different GO classes that were enriched in at least one of the five cluster groups. All treatments were sorted in the same order as for PTU 5 dpf, to enable to visualize the change of response pattern between the time points and compounds.

**Figure 4 ijms-20-01543-f004:**
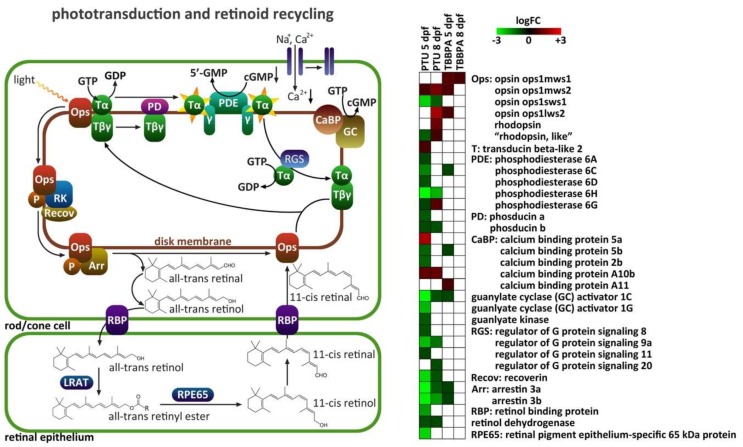
Overview of the effects of propylthiouracil (PTU) or tetrabromobisphenol-A (TBBPA) exposure on transcript levels of components of the phototransduction and retinoid recycling pathways. Left: detailed pathways in rod/cone cells and the retinal epithelium; Right: heatmap showing the transcript level changes of components of the pathways. LRAT: lecithin retinol acyltransferase. All other abbreviations are explained in the heatmap on the right. (Figure adapted from Houbrechts [26]).

**Table 1 ijms-20-01543-t001:** Test set-up for THDC exposure with subsequent recovery phase.

	0 dpf	1 dpf	2 dpf	3 dpf	4 dpf	5 dpf	6 dpf	7 dpf	8 dpf
**Control**	W	W	W	W	W	X			
**Control recovery**	W	W	W	W	W	W	W	W	X
**PTU (350 mg/L)**	C	C	C	C	C	X			
**PTU recovery**	C	C	C	C	C	W	W	W	X
**TBBPA (200 µg/L)**	C	C	C	C	C	X			
**TBBPA recovery**	C	C	C	C	C	W	W	W	X

W: clean water, C: chemical exposure, X: sampling, dpf: days post fertilization; each treatment group (rows) was run in 3 biological replicates (*n* = 24 larvae each).

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
