# Peer review of "Thyroid Hormone Disruptors Interfere with Molecular Pathways of Eye Development and Function in Zebrafish"

_ijms, 2019, doi:10.3390/ijms20071543_

Round 1

Reviewer 1 Report

Baumann and colleagues studied the effects of thyroid hormone disruptors on molecular  pathways of eye development and function in zebrafish. In vitro experimentation has been performed using 1 concentration of either propylthiouracil (PTU), a thyroid hormone inhibitor, or tetrabromobisphenol-A (TBBPA), from 0 - 5 days post fertilization (dpf). In addition, the authors have also investigated the reversibility of effects after recovery in clean water for three days. Affected transcriptional patterns have been assessed by full genome microarray analyses of RNA isolated from eye tissue.

The text is accurately written, the work well conducted and the techniques applied are reliable. The results are interesting but add very little (substantially confirm) previous literature on the effects of THDCs on eye development and function in fish. 

Specific comments:

Introduction. I would suggest adding few more lines about TBBPA accumulation on aquatic species or its environmental concentration. In other words, is the TBBPA concentration somehow environmentally justified?

Materials and Methods.

line 385. The authors stated the nominal concentrations, but they must write the final one in the medium.

line 404. Specify how many breeding pairs.

line 414. "three replicate plates of each treatment were sampled", I do not believe that 3 replicates are enough for a good statistical design, thus authors must provide the rational of using such numbers. Furthermore, what about biological replication!!??

line 417What is the rational to use 3 days (i.e. until 8 dpf) as recovery period!?

Results and Discussion.

I perfectly understand the goals of the present paper (i.e. pathway discovery) but, in my opinion, it looks too much descriptive. I strongly believe that it would be useful to perform follow-up qPCR experiments which are basic for giving you a proper indication if the dysregulation found by microarray may be of biological relevance. In my opinion, the key point is exactly the biological relevance of the observed findings. To increase the strength of the manuscript, I would suggest to further investigate the most important array results (at least the effects on vision-related transcripts at 5 dpf) by qPCR, to get a better idea of the possible biological relevance.

Overall, I found poorly discussed the most surprising finding of the study that is the over 10-times higher number of differentially expressed transcripts by PTU! Authors should comment on this.

Authors show activation of substantial repair and regeneration processes, but also point out the lack of a fully accomplished recovery process. I think that these conclusions are not well supported when evaluating the available information.

Minor comments

Line 13. Check the number relative to the affiliation.

Line 92. Remove "2. Results".

Line 290. Please add a dot after "[44].

Author Response

We would like to thank the reviewers for their helpful suggestions, the time they invested for reviewing and for critically reading this manuscript. They have allowed us to clarify and elaborate on certain aspects of the manuscript that required attention. We hope that these modifications comply with the reviewer’s remarks and help to improve the quality of the manuscript. Modifications in the manuscript are marked in yellow color.

The following specific comments of the reviewers were considered for the modifications in the manuscript (answers in bold letters):

Reviewer 1:

Introduction. I would suggest adding few more lines about TBBPA accumulation on aquatic species or its environmental concentration. In other words, is the TBBPA concentration somehow environmentally justified?

We have extended the paragraph that mentions the environmental concentrations of TBBPA (see lines 60-64). Moreover, we compare our exposure concentrations to those used in other studies (see lines 78-81).

line 385. The authors stated the nominal concentrations, but they must write the final one in the medium.

We did not perform chemical analyses of the exposure solutions. As mentioned in lines 442-444, we ensured constant chemical concentration levels and good water quality by preparing fresh test solutions every day and by fully renewing the exposure solution in the embryo plates every day. Additionally, exposure plates were pre-incubated to ensure saturation (line 440).

line 404. Specify how many breeding pairs.

We have added the information to M&M (see line 434).

line 414. "three replicate plates of each treatment were sampled", I do not believe that 3 replicates are enough for a good statistical design, thus authors must provide the rational of using such numbers. Furthermore, what about biological replication!!??

Three biological replicates were used. The way the experiment was set up ensures that there is biological variation among these three replicates, in addition to technical variation. Many previous studies (e.g. Martyniuk et al. 2007, Aquatic Toxicology, Vol. 84, Iss. 1, P. 38-49; Mathavan et a. 2005, PLOS Genetics, 1 (2): e29; Vergauwen et al. 2010, CBP, Part A 157, 149-157) have shown that the use of 3 biological replicates (based on pooled samples) is appropriate to detect consistent molecular responses using microarrays. Higher numbers of replicates would of course allow to detect more subtle responses but would also result in a disproportionately higher costs of the microarray analysis. We revised the paragraph that describes the experimental set-up and specified the description of the replicates (see lines 438/439, 444-450).

line 417. What is the rational to use 3 days (i.e. until 8 dpf) as recovery period!?

In our study by Houbrechts et al. (2016), we observed recovery from altered eye development at 7 dpf that was induced by dio-knockdown (mentioned in the discussion). This can be explained by the chosen method for the gene knockdown with morpholinos, that is known to be not fully persistent over longer periods. Based on this, we chose our exposure scenario with 5+3 days. Moreover, this took the short half-lifes (few hours) of our exposure compounds into account. We added this information to M&M (see lines 451-456).

I perfectly understand the goals of the present paper (i.e. pathway discovery) but, in my opinion, it looks too much descriptive. I strongly believe that it would be useful to perform follow-up qPCR experiments which are basic for giving you a proper indication if the dysregulation found by microarray may be of biological relevance. In my opinion, the key point is exactly the biological relevance of the observed findings. To increase the strength of the manuscript, I would suggest to further investigate the most important array results (at least the effects on vision-related transcripts at 5 dpf) by qPCR, to get a better idea of the possible biological relevance.

In our opinion, it is not really the purpose of qPCR confirmation to give an indication of the biological relevance of the microarray data. For this, it is important to add data from higher levels of biological organization, which is what we show in our first study and build on in the present one. Confirmation of single genes should be performed when the role of specific transcriptional changes at gene level are discussed. We decided against further qPCR analyses, as our conclusions are entirely process/pathway-based and not based on individual genes or the fold-changes of transcriptional regulation. We were interested in affected biological processes and therefore, think that further analyses of single genes would not add specific value to our study and will not change the conclusions we could draw from our pathway analyses. We added a respective sentence to the introduction (see lines 98-100).

Overall, I found poorly discussed the most surprising finding of the study that is the over 10-times higher number of differentially expressed transcripts by PTU! Authors should comment on this.

We have added a paragraph to the discussion (see lines 233-241).

Authors show activation of substantial repair and regeneration processes, but also point out the lack of a fully accomplished recovery process. I think that these conclusions are not well supported when evaluating the available information.

While this was not intended, we understand that parts of this section appeared to go beyond the data that was presented in the manuscript. We changed the text at the end of the discussion to clarify this (lines 394-402).

Line 13. Check the number relative to the affiliation.

Line 92. Remove "2. Results".

Line 290. Please add a dot after "[44].

Thank you for pointing out those mistakes. We have corrected them.

Reviewer 2 Report

The article of Baumann et al. brings important contributions to understand the molecular pathways in the eye of exposed zebrafish to THDCs. The study uses two model compounds that have different molecular initiating events, but lead to similar phenotypic adverse effects in eye development.

The major novelty that brings this paper is that uses tissue-specific analyses of isolated eye tissue to avoid dilution of gene expression signals of the target organ by other tissues. I have no major concerns to technical aspects of the work and would recommend acceptance of the manuscript after some minor revision.

Omics-based methods are increasingly used in current ecotoxicology and may be used for identifying mode of actions or adverse outcomes pathways. The authors show that PTU had a much stronger impact on transcriptional changes in the eyes of exposed fish than TBBPA, even though morphological and physiological changes in the eyes were comparable in a previous study of the same authors. It would be helpful to anchor the exposure concentration to a quantitative measurable phenotypic or physiological effect (for instance an ECx). That might help us to interpret such differences on transcriptional level. On the other hand, it could also be discussed along the text, the fact that using a concentration-resolved experiment should offer a better understanding of response patterns and establishment of the causal relationship between gene expression and adverse outcome pathway.

Specific comments:

Line 92: Remove duplication: “2.Results”

Results section: It would help the reader if in the result section, at least concentrations and rational for using these concentrations is explained before presenting results. That’s because material and methods are located at the end.

Results figure 4: please provide abbreviations of the adapted figure from Houbrechts.

Line 385: provide city, country of provider Sigma-Aldrich.

Line 392: provide reference to Westerfield, M. book instead of zfin.org

Line 413-14: Was the chemical exposure performed using the same batch of eggs? Are these 6 replicates, biological (different batch of eggs) or technical replicates?

Line 461: please provide citation to R package ‘arrayQualityMetrics’

Line 469: please provide citation to R package ‘Limma’

Supplemental information: please indicate label of the y axis in figures S1-5. What does mean “logFC” in the x-axis?; please provide that in the description of the figures.

Author Response

We would like to thank the reviewers for their helpful suggestions, the time they invested for reviewing and for critically reading this manuscript. They have allowed us to clarify and elaborate on certain aspects of the manuscript that required attention. We hope that these modifications comply with the reviewer’s remarks and help to improve the quality of the manuscript. Modifications in the manuscript are marked in yellow color.

The following specific comments of the reviewers were considered for the modifications in the manuscript (answers in bold letters):

Reviewer 2:

The authors show that PTU had a much stronger impact on transcriptional changes in the eyes of exposed fish than TBBPA, even though morphological and physiological changes in the eyes were comparable in a previous study of the same authors. It would be helpful to anchor the exposure concentration to a quantitative measurable phenotypic or physiological effect (for instance an ECx). That might help us to interpret such differences on transcriptional level. On the other hand, it could also be discussed along the text, the fact that using a concentration-resolved experiment should offer a better understanding of response patterns and establishment of the causal relationship between gene expression and adverse outcome pathway.

We fully agree with the reviewer and have added a paragraph that explains which quantifiable endpoints were assessed in our first study and that the chosen exposure concentrations were based on similar effect levels, i.e. LOECs for adverse effects on eye morphology and physiology (see lines 414-419). The aim of the present study was to identify the molecular pathways involved in those observations and, therefore, we did not perform further experiments with multiple concentrations (as in our previous one). 

Line 92: Remove duplication: “2.Results”

Done.

Results section: It would help the reader if in the result section, at least concentrations and rational for using these concentrations is explained before presenting results. That’s because material and methods are located at the end.

We fully agree with the reviewer. However, we decided to add this information to the introduction, instead of the result section, as we believe it fits better to have such an explanation there (see lines 83-86).

Results figure 4: please provide abbreviations of the adapted figure from Houbrechts.

The abbreviations are given in the figure itself. We added this information to the figure caption (line 217).

Line 385: provide city, country of provider Sigma-Aldrich.

Done.

Line 392: provide reference to Westerfield, M. book instead of zfin.org

Done.

Line 413-14: Was the chemical exposure performed using the same batch of eggs? Are these 6 replicates, biological (different batch of eggs) or technical replicates?

We have modified this paragraph according to the requests of both reviewers. We hope that the experimental set-up is clearer now (see lines 438 ff).

Line 461: please provide citation to R package ‘arrayQualityMetrics’

Line 469: please provide citation to R package ‘Limma’

We have added a reference for both.

Supplemental information: please indicate label of the y axis in figures S1-5. What does mean “logFC” in the x-axis?; please provide that in the description of the figures.

Done.

Round 2

Reviewer 1 Report

I sincerely appreciate the authors' reply to my comments and I think that the revised version of the manuscript has been substantially improved.

Please, check again the affiliation list: I cannot find the number 5!